# Experimental Study on Mix Proportion Parameter Optimization of Cement Anchoring Material

**DOI:** 10.3390/ma13010137

**Published:** 2019-12-28

**Authors:** Jinzhang Jia, Bin Li, Fei Liu

**Affiliations:** 1College of Safety Science and Engineering, Liaoning Technical University, Fuxin 123000, China; 2Key Laboratory of Mine Power Disaster and Prevention of Ministry of Education, Huludao 125105, China; 3China Coal Technology Engineering Group Shenyang Research Institute, Shenyang 110016, China

**Keywords:** orthogonal design method, anchoring material ratio, mechanical property index, multiple linear regression analysis, safety of anchorage system

## Abstract

To prevent major deformation of surrounding rock and improve the stability of unstable rock mass, this study optimized the ratio of cement anchoring material based on the orthogonal design method and rock mechanics tests. Using the six influencing factors of coal ash content, sodium silicate content, ettringite content, naphthalene sulfonate content, water–cement ratio, and sand–cement ratio, a total of 18 sets of material matching schemes was designed. Compressive strength and pull-out force tests were carried out. The results are as follows: the compressive strength of the test piece between 43 and 55 MPa; the bond stress between the bolt and the anchoring agent range is from 103 to 136 kN; and the adhesion stress between the anchoring agent and the rock masses range is from 76 to 112 kN. The main order and degree of influence of various factors affecting the bond stress and the adhesion stress of the anchoring material were determined by range analysis and analysis of variance. The results showed that the amount of coal ash had a clear controlling effect on the bond stress, and the water–cement ratio, sand–cement ratio, and sodium silicate dosage had a significant influence on the bond stress. Moreover, the amount of sodium silicate had the greatest influence on the adhesion stress, and the water–cement ratio had the second largest influence on the adhesion stress. The amount of naphthalene sulfonate had the least influence on the adhesion stress. The above experimental data and results were subjected to multiple linear regression analyses, and the empirical equations of the mechanical indexes of the test pieces and the proportion parameters of the anchoring materials were obtained to guide the engineering support design. The engineering application showed that the optimized anchorage material could be applied to further improve the safety of the surrounding rock anchoring system.

## 1. Introduction

Since the first use of anchors for underground roadway support in the early 19th century, anchor bolt supports have been used for more than 100 years [1,2]. As a kind of active support technology with high support efficiency and low cost, anchor bolt support could fully improve the self-stability of rock masses and effectively control the deformation of rock mass engineering [3]. Moreover, it could ensure the safety of construction and the stability of engineering, and has been widely used in various roadway projects in various countries [4]. In recent years, with the continuous increase in the depth of mining, it has been difficult to support roadways because of high ground stress, strong mining, broken surrounding rock, and soft rock [5,6]. Therefore, the requirements for anchor bolt supports have also increased. In response to the problem of roadway support, scholars from various countries have conducted numerous studies and achieved many important results.

Zhou et al. [7] found that the incorporation of steel beads into the mortar anchoring agent can greatly increase the elastic modulus, but the uniaxial compressive strength decreases by about 10%. Kilic et al. [8,9,10] showed that anchoring agent interface disruption was the main mode of anchorage failure. Their tests showed that if the anchoring length is short and the surrounding rock is hard, parallel to shear failure of the anchoring agent occurs along the tip of the anchor rib. At the same time, they concluded that the transverse rib spacing of the anchor had a considerable influence on the load transfer of the anchoring system. Based on the shear–lag model, Cai et al. [11,12] proposed a theoretical analysis model considering the interaction between anchor, anchoring agent, and surrounding rock, and studied the mechanical distribution of the anchoring interface in the pull-out test. Hyett et al. [13,14,15] determined two failure modes in cement grouting anchor cables based on the pull-out test in laboratory and field tests. One was the radial splitting of the concrete surrounding the anchor cable, and the other was the shear failure of the interface between the anchor cable and the concrete. Cao et al. [16,17] analyzed the failure mode and failure process of the anchorage section. It was believed that the addition of steel aggregate to the resin anchoring agent could change the direct shear failure mode of the original anchoring agent, thereby improving the shear strength of the anchoring agent during the failure of the anchoring section. Benmokrane et al. [18] performed a pull-out test of rebars anchored by six different types of cement grouting, and the results showed that the correlation between the uniaxial compressive strength of the anchoring agent material and the peak strength of the bond was very weak. Mitsui et al. [19] studied the mechanical properties of the coarse aggregate–cement interface and used the push-out test method to analyze the interfacial properties of aggregate-cement paste: shear stiffness, bond strength, friction strength, and fracture energy. Caliskan et al. [20] conducted a similar study as Mitsui et al., but using different types of coarse aggregates. The findings of Caliskan et al. show that the addition of silica fume to the cement paste can enhance the bond strength of the interface, and the interfacial bond strength has a significant size effect, which decreases with the increase of the coarse aggregate size.

The above researchers used the combination of theory and experiment to explore the mechanisms of the mechanical properties of the anchoring interface. It was revealed that for any kind of anchoring system, the stress transmission mode was transmitted from the anchor body of the anchoring agent, and then transferred to the rock mass by the anchoring agent, involving a complicated transfer process of three media and two interfaces. The bonding between the rod and the anchoring agent, and the bonding between the anchoring agent and the rock mass were the main factors determining the ultimate bearing capacity of the anchoring system. Once one of the two interfaces failed, the entire anchoring system would be destroyed, resulting in the destruction of the entire structure [21]. Therefore, by improving the bonding effect of anchoring agent at the failure interface, the reinforcement effect of the whole anchoring system would be greatly improved. Based on this, this study took the rock mass support of the Xincheng Gold Mine as an example. The study focused on the improvement on the mechanical properties of the anchoring system by optimizing anchoring agent material ratio. The above research results can also be applied to other similar mine support engineering fields.

## 2. Background

### 2.1. Geological Conditions

The survey site selected for this test was Xincheng Gold Mine. Xincheng Gold Mine is located in the southeastern part of China and is mainly composed of I# and V# ore bodies. The V# ore body was distributed over the 165–188 line below the middle section of −430 m. The ore body was layered, and the thickness was generally between 10 and 30 m, with an average of 24.4 m. The horizontal ground stress was 31.38 MPa, and the vertical ground stress was 17.35 MPa. The roof rock of the ore body was mainly sericitolite, and the bottom plate was granodiorite. The relative stability of the ore body and the top and bottom rock mass was poor, the engineering geological conditions of the local fracture zone or altered rock were also poor, and the fracture structure development was a prominent feature. Figure 1 shows a three-dimensional view of the V# ore body.

### 2.2. Roadway Support and Deformation Characteristics

The return roadway of the V# ore body of Xincheng Gold Mine is a 3.5 m × 3.3 m three-heart arch (Figure 2). The basic support form of the roadway is “bolt + U-shaped steel + metal mesh”. The anchor rod is made of ordinary reinforced cement anchor with Φ25 mm and L = 2500 mm. The spacing between the anchors is 800 mm × 800 mm, and the length of the anchoring section is 1.8 m. The metal mesh comprised a double-layer welded wire mesh and a bolt to form an anchor spray net support. A U-shaped steel support was used in the stress concentration area of the rock mass. The metal mesh is Φ8 mm, the mesh is 120 mm × 120 mm, the specification is 1500 mm × 1000 mm, and the thickness of the shotcrete is 50–80 mm. As a result of the deep burial of the roadway, the deformation of the roadway was severe under the action of high stress. There were many phenomena such as peeling of the skin, sinking of the roof, deformation of the bottom drum, failure of the original anchor bolt support, and tilting of the rock. The monitoring results of surrounding rock deformation data showed that the maximum deformation of the two sides can reach 450 mm and the maximum deformation of the roof floor could reach 600 mm within 90 days after excavation. This shows that the support effect of the roadway was not satisfied.

## 3. Anchoring Agent Material Ratio Orthogonal Design Method

### 3.1. Principles of the Orthogonal Test

The orthogonal test method selects representative and typical test points from a large number of test points in the multi-factor optimization experiment by using the principles of mathematical statistics and orthogonality. Orthogonal tables are used to scientifically arrange and analyze the effects of multiple factors to achieve optimal test results from as few trials as possible [22,23]. In the orthogonal test, the test results are analyzed by the range analysis method and the variance analysis method. Through the analysis of the extreme difference between the test data, the primary and secondary sequence of the influencing factors can be arranged intuitively; the variance between the experimental data can be used to obtain the experimental error, to make up for the shortcomings of the range analysis, and to improve the accuracy of the test [24].

(1) The principle of range analysis
(1)$$R_{j}=\max\left(k_{ij}\right)-\min(k_{ij})$$
(2)$$k_{ij}=K_{ij}/s$$
where *R_j_* is the range of the *j* column, (*k_ij_*) is the arithmetic mean of the test results corresponding to the *i* level in the *j* column of the orthogonal table, *K_ij_* is the sum of the corresponding test indicators when the element on the *j* column in the orthogonal table takes the *i* level, and *s* is the number of times the element *i* appears horizontally on the *j* column in the orthogonal table.

(2) The principle of variance analysis

To compare the difference between the index values of the factors, the factor A is taken as an example to calculate the sum of squared deviations of the factors:(3)$$S_{A}={\sum_{i=1}^{r}\sum_{j=1}^{a}\left(\bar{y_{i}}-\bar{y}\right)}^{2}$$
where *S_A_* is the one-way analysis of variance (ANOVA), $\bar{y_{i}}$ is the average of the *i* level test results, and $\bar{y}$ is the mean of all test results.
(4)$$\left\{ \begin{array}{l} \bar{y_{i}}=\frac{1}{a}\sum_{i=1}^{r}y_{ij}\\ \bar{y}=\frac{1}{n}\sum_{k=1}^{n}y_{k} \end{array}\right.$$
where *y_ij_* is the *j* test result of the *i* level of a factor (*i* = 1, 2, …., *r*; *j* = 1, 2, …, *a*), and *y_k_* is the total test result value (*k* = 1, 2, …, *n*).
(5)$$\sum_{i=1}^{r}\sum_{j=1}^{a}y_{ij}=\sum_{k=1}^{n}y_{k}$$
(6)$$S_{A}=\frac{1}{a}{\sum_{i=1}^{r}\left(\sum_{j=1}^{a}y_{ij}\right)}^{2}-\frac{1}{n}{\left(\sum_{i=1}^{r}\sum_{j=1}^{a}y_{ij}\right)}^{2}=\frac{1}{a}\sum_{i=1}^{r}K_{i}^{2}-\frac{1}{n}{\left(\sum_{k=1}^{n}y_{k}\right)}^{2}$$
where *a* and *n* is the number of trials, and *K_i_* is the sum of the results of the *i* level *a* test of a factor. 

The sum of squared errors was caused by the random error of the sample and represents the difference between the sample value and the sample mean at the level A*_i_*. It can be expressed as: (7)$$S_{E}=S_{T}-S_{Y}$$
where *S_E_* is the sum of squared errors, *S_T_* is the sum of squared total deviations, and *S_Y_* is the sum of squared deviations representing all factors.

In the analysis of variance, the size of the *F* value can be compared to determine whether there is a significant difference between the groups. The calculation method is as follows:(8)$$F=\frac{S_{Y}}{f_{Y}}/\frac{S_{E}}{f_{E}}$$
where *f_Y_* is the degree of freedom of each factor, and *f_E_* is the degree of freedom of test error.

The test level *a* is given, and the *F* distribution table is checked by *F**a* (*f_Y_*, *f_E_*). If *F* > *Fa* (*f_Y_*, *f_E_*), this means that the influence of this factor of the test results has reached a significant level.

### 3.2. Orthogonal Design Scheme

Ordinary Portland cement (42.5 grade) was selected as the main agent of this test. Its density is 3.06 g/cm^3^ and the specific surface area is 350 m^2^/ kg. Its initial setting time is 183 min and final setting time is 237 min. Table 1 shows its chemical composition. Then, the ettringite, sodium silicate, naphthalene sulfonate, and coal ash were selected as anchoring material admixtures. The specific properties of the admixtures are shown in Table 2.

The amount of coal ash, the amount of sodium silicate, the amount of ettringite, the amount of naphthalene sulfonate, the water–cement ratio, and sand–cement ratio of the anchoring agent material were the six factors of the orthogonal design, and each factor was set to three-level control. In combination with the actual needs of the experimental design of this paper, the orthogonal level design table of three levels and seven factors was used. In practice, an empty column was reserved for error analysis. A total of six factors were considered, as shown in Table 3. The specific settings for the orthogonal design scheme L18 (3^7^), including a total of 18 experimental mix proportion schemes, are shown in Table 4.

## 4. Anchoring Agent Material Ratio Test Process

This test was mainly conducted to study the effect of the inherent properties of cement anchoring materials on the strengthening of anchoring systems. The compressive strength of the anchoring agent material, the adhesion stress between the anchoring agent material and the anchor rod, and the bond stress between the anchoring agent material and the rock mass were selected as test indicators. According to the material ratio design scheme of Table 4, three sets of different index tests were carried out. The test procedure was as follows:

(1) Compressive strength test: according to the anchoring agent material ratio scheme in the orthogonal design, 18 different compressive strength test pieces with different material ratios were prepared. The size of each test piece was 70.7 mm × 70.7 mm × 70.7 mm. Each test piece had a smooth surface without cracks.

As shown in Figure 3, all of the test pieces were dried to a constant mass, cooled to room temperature, and placed in a curing box for 28 days. After the test piece reached the maintenance period, it was placed on the pressure plate of the test machine. As shown in Figure 4, the center of the pressure plate of the test press was coincident with the center of the test piece. Then the test machine was started, the speed of compression is 10 mm/min, the parameter values were set, and the peak value was recorded. The results are shown in Table 5.

(2) Bond stress test: a total of 18 hollow steel pipes were prepared with a length of 0.6 m and a diameter of 150 mm as the simulated anchor holes. The corresponding number of mining cement anchors were selected, each with a length of 1.5 m and a diameter of 22 mm, and the anchoring materials with different mixing ratios were then inserted. Once the bottom of the test piece was leak-proof, it was placed in a flat and ventilated place for 28 days, as shown in Figure 5. After the test piece was shaped, the drawing test was performed.

After the puller proofreading was completed, the test piece was placed in a horizontal plane to stabilize it. First the washer was put into the bolt above the test piece, and then the hydraulic support end of the puller was placed into the bolt above the washer, and fixed with a single hole anchorage device. After the test piece and the puller were connected and fixed, the pull test was performed, as shown in Figure 6. This test utilized a manual hydraulic puller, speed of pulling is 15 mm/min. Therefore, it was possible to observe the degree at any time during the pressurization process, and the maximum bond stress was obtained after the bolt was pulled out, the results are shown in Table 5. After that, the test piece and the external part of the puller were disassembled for the next set of tests.

(3) Adhesion stress test: the rock blocks of certain geometrical dimensions were recovered on site and placed flat on the ground. The rock was drilled vertically using an impact hammer (hole diameter 72 mm and hole depth 400 mm), and the borehole was cleaned with high pressure air at a pressure on 1 MPa, as shown in Figure 7. Then, different ratios of cement paste materials were poured into the anchor holes, and the mixture was uniformly stirred by iron brazing. The anchor rod was inserted into the hole, the circular tray with the same diameter as the hole was arranged on the anchor rod, and 18 different material mixture ratio test pieces were prepared according to the orthogonal design scheme, as shown in Figure 8. After curing for 28 days under natural conditions, the adhesion stress test was carried out, as shown in Figure 9. The speed of pulling was 15 mm/min. After all the anchor rods were pulled out, the loading force value was recorded. The test results are shown in Table 5.

## 5. Anchoring Agent Material Orthogonal Test Result

Three mechanics indexes of anchoring materials were tested using the orthogonal design method in this study: compressive strength, adhesion stress, and bond stress. The test results are shown in Table 5. In this section, the two indicators of adhesion stress and bond stress were analyzed by the range and variance analysis. According to the requirements of the “Technical Specification for ground anchors”, the compressive strength of the grouting body of the bolt anchorage section was not less than 30 MPa [25]. In the test, the compressive strength of 18 kinds of anchoring material samples with different ratios was between 43 and 55 MPa after 28 days, all of which were greater than 30 MPa, and thus the optimized ratio analysis was not carried out.

### 5.1. Analysis of the Bond Stress in the Orthogonal Test of the Anchoring Agent Material

The test results of the bond stress test were substituted for Formulas (1) and (2), and the average and the range were obtained (Table 6). The three mean values in the table represent the average bond stress strengths of the six factors at three different levels. The level that maximizes the mean among each factor was selected as the optimal level. Therefore, the primary and secondary relationships of the influence of various factors on the bond stress were the coal ash content, sand–cement ratio, sodium silicate content, water–cement ratio, naphthalene sulfonate content, and ettringite content. To more intuitively analyze the influence of various factors of the bond stress, the bond stress analysis in Table 6 was used as a factor to draw a visual analysis diagram (Figure 10). The optimal combination of factor level was as follows: coal ash content of 12%, sodium silicate content of 5%, ettringite content of 6%, naphthalene sulfonate content of 0.5%, water–cement ratio of 0.4, and sand–cement ratio of 1.5.

Figure 10a shows that the bond stress increased with the increase of the amount of coal ash. The packing action of the reaction products of coal ash and cement paste decreased the macroporosity and capillary porosity in the cement stone while increasing the gel pores. The large change in the pore size distribution (large holes reduced and small holes increased) made the structure more compact and uniform and thereby the contact area between the paste and the anchor was increased [26]. The frictional resistance to the anchor and the paste was greatly improved, thereby effectively improving the bond stress between the anchor and the paste.

Figure 10b shows that the size of the bond stress increased as the amount of sodium silicate was increased. This was because sodium silicate can generate a gel reaction with metals or metal oxides. The alkali in the sodium silicate is seized by the metal ions, which causes the sodium silicate to lose water to form silica gel, thereby increasing the strength of the bond stress between the anchor and the paste [27].

Figure 10c shows that the size of the bond stress changed little with the increase in ettringite. Adding the proper amount of ettringite to expand the volume of the paste can increase the mechanical interaction force between the contact surfaces, which could improve the bond stress [28].

Figure 10d shows that the size of the bond stress decreased initially and then increased owing to the increase in the naphthalene sulfonate content; however, the overall effect was not large. By reducing the amount of water used, the compressive strength of the anchorage material will be greatly improved.

Figure 10e shows that the size of the bond stress decreased as the water–cement ratio increased, but this does not mean that the water cement ratio could be reduced to increase the bond stress. If the water–cement ratio was too small, the cement paste would be difficult to grout, the hydration reaction would be incomplete, and the cement could not be evenly stirred [29].

Figure 10f shows that the size of the bond stress first increased and then decreased as the sand–cement ratio increased. This means that the sand–cement ratio had an optimal ratio in terms of bond stress. At the optimum ratio, the bond stress of the anchoring agent material was the highest, and the optimum effect was not obtained when the ratio was less than or greater than this ratio.

The test results of the bond stress test were substituted for Formulas (3)–(8) for analysis of variance, and the confidence values were taken as 90%, 95%, and 99% (Table 7).

According to the F ratio of the analysis of the variance of the bond stress, the order for the influence of each factor of the bond stress was: coal ash content, sand–cement ratio, sodium silicate content, water–cement ratio, naphthalene sulfonate content, and ettringite content, which was consistent with the results of the range analysis. For the strength of the bond stress, the amount of coal ash reached a significance level of 90%, 95%, and 99% confidence. The sodium silicate content, sand–cement ratio, and water–cement ratio reached a significant level when the confidence was 90% and 95%. This shows that these four factors had a significant impact on the bond stress. The naphthalene sulfonate content reached a significant level at a confidence level of 90%, indicating that it had a certain influence on the bond stress. Although the ettringite content did not reach a significant level, its squared deviation was larger than the square of the deviation from the error. This shows that the all results of the orthogonal test were reasonable.

### 5.2. Analysis of Adhesion Stress of the Orthogonal Test of the Anchoring Agent Materials

The test results of the adhesion stress test were substituted for Formulas (1) and (2), and the average value and range were determined (Table 8). The primary and secondary relationships of the influence of each factor on the adhesion stress were analyzed. The order was sodium silicate content, water–cement ratio, coal ash content, ettringite content, sand–cement ratio, and naphthalene sulfonate content. Based on the analysis results presented in Table 8, an intuitive analysis of the influence of various factors of the adhesion stress was conducted, as shown in Figure 11. The optimal combination was as follows: coal ash content of 12%, sodium silicate content of 5%, ettringite content of 6%, naphthalene sulfonate content of 0.5%, water–cement ratio of 0.4, and sand–cement ratio of 2.

Figure 11a shows that the adhesion stress increased as a result of the increase of the coal ash content. Since the coal ash admixture contained active SiO_2_, it reacted with the hydration product of the cement and the free lime to form a C–S–H gel [30], which could improve the compactness of the anchoring agent and the adhesion stress with the contact surface of the rock mass.

Figure 11b shows that the adhesion stress increased as a result of the increase in the sodium silicate content. When sodium silicate was added, it immediately reacted with the Ca(OH)_2_ produced during cement hydration to form a large amount of calcium silicate gels. As the reaction progressed, more and more colloids were formed, which helped to improve the adhesion stress between the contact faces [27].

Figure 11c shows that the adhesion stress increased as a result of the increase in the ettringite content. As an expansion agent, ettringite can effectively increase the volume of the anchoring agent in the anchor hole. Therefore, it can enhance the friction stress and the average shear stress between the two contact surfaces, which improves the adhesion stress between the anchoring agent and the rock body contact surface.

Figure 11d shows that the adhesion stress surface first decreased and then increased from the increase in the naphthalene sulfonate content, but had no significant effect on the adhesion stress overall. Its role was to reduce water consumption and improves the durability of the anchoring agent.

Figure 11e shows that the adhesion stress increased from the decreased in the water–cement ratio. This is because the higher the water-cement ratio is, the higher the water content of the cement paste is and the more voids in it are, which leads to the decrease of the bond stress [31].

Figure 11f shows that the adhesion stress increased from the increase in the sand–cement ratio. The higher the sand–cement content, the greater the bulk expansion when the anchoring agent interface deforms. As a result of the limitation of the anchor hole, the frictional resistance would become larger and larger, and thus the adhesion would be greater [32]. 

The test results of the adhesion stress test were substituted for Formulas (3)–(8) for analysis of variance, and the confidence values were taken as 90%, 95%, and 99% (Table 9).

According to the F ratio of the analysis of the variance in the adhesion stress, the order for the influence of each factor of the adhesion stress was as follows: sodium silicate content, water–cement ratio, coal ash content, ettringite content, sand–cement ratio, and naphthalene sulfonate content, which was consistent with the results of the range analysis. For the adhesion stress, the sodium silicate content and water–cement ratio reached a significance level of 90%, 95%, and 99% confidence. The coal ash content, ettringite content, and sand–cement ratio reached a significant level when the confidence was 90% and 95%. This indicates that these five factors had clear effects on the adhesion stress, although the content of naphthalene sulfonate did not reach a significant level. However, the square of the deviation was larger than the sum of the squares of the error deviations, which indicates that all the results of the orthogonal test were reasonable.

### 5.3. Optimization Analysis of the Anchoring Agent Material Ratio

Based on the analysis of the influence of each factors on the bond stress and the adhesion stress respectively, the influence of each factor on the two test indicators were analyzed comprehensively, and the optimal combination of each factor and level given. The optimal combination is shown in Figure 12.

Figure 12a shows that the bond stress and the adhesion stress increased with the increase in the coal ash content and 12% was taken as as the optimum content.

Based on Figure 12b, it could be concluded that the bond stress and the adhesion stress increased from the increase in the sodium silicate content and 5% was taken as as the optimum content.

Figure 12c reveals that the bond stress first increased and then decreased with the increase of ettringite content, but the overall effect was not significant. The adhesion stress had a significant influence on the increase of the ettringite content and 6% was taken as the optimum content.

Figure 12d shows that the bond stress and the adhesion stress were not significantly changed by changes to the naphthalene sulfonate content. Considering that proper reduction of water consumption could impove compressive strength, 1.5% was taken as the optimum naphthalene sulfonate content.

Figure 12e shows that both of the mechanical characteristics the bond stress and the adhesion stress decreased as the water–cement ratio increased. This is consistent with the law that practically all the mechanical characteristics of the cement paste deteriorates with the increase of the water-cement ratio. Therefore, 0.4 was taken as the optimum water–cement ratio.

Based on Figure 12f, it could be concluded that the bond stress increased slowly with the sand–cement ratio between 1 and 1.5, and the sand–cement ratio decreased rapidly when the ratio was between 1.5 and 2. The adhesion stress increased rapidly with the sand–cement ratio between 1 and 1.5, and the sand–cement ratio increased slowly between 1.5 and 2. Therefore, 1.5 was taken as the optimum sand–cement ratio.

Based on the above analysis, the optimal mix ratio of the anchoring agent material was obtained, as shown in Table 10. Since the optimal combination of anchorage material parameters did not appear in the previous 18 sets of tests, the latest mix ratio was tested to verify its optimization. The test results are shown in Table 11. After parameter optimization, the bond stress and adhesion stress were better than in the previous 18 sets of test results.

### 5.4. Multiple Linear Regression Analysis of the Anchoring Agent Material Ratio

Multiple linear regression analysis was used in this study. Through regression analysis, empirical formulas can be given to guide practical applications [33,34,35].

(1) Multiple linear regression analysis model
(9)$$y=b_{0}+b_{1}x_{i1}+b_{2}x_{i2}+...+b_{m-1}x_{i,m-1},\;i=1,2...n$$

The coefficients and constant terms in the model were obtained by the least squares method, so that:(10)$$Y={\left(\begin{array}{l} y_{1}\\ y_{2}\\ \;\vdots \\ y_{n} \end{array}\right)}_{n\times 1},\;X=\left(\begin{array}{l} 1\;x_{11\;}x_{12}\;\cdots \;x_{1,m-1}\\ 1\;x_{21}\;x_{22}\;\cdots x_{2,m-1}\\ \;\vdots \;\vdots \;\vdots \;\vdots \;\\ 1\;x_{n1}\;x_{n2}\;\cdots x_{n,m-1} \end{array}\right),\;\beta ={\left(\begin{array}{l} b_{0}\\ b_{1}\\ \;\vdots \\ b_{m-1} \end{array}\right)}_{m\times 1}$$

The regression model is:(11)Y=Xβ
where *Y* is the observation vector, *X* is the design matrix, and *β* is an unknown parameter vector to be estimated. The point at which *β* could be obtained by calculation was estimated as:(12)$$\beta =\left(\begin{array}{l} b_{0}\\ b_{1}\\ \;\vdots \\ b_{m-1} \end{array}\right)={\left(X^{T}X\right)}^{-1}X^{T}Y$$

Visual analysis of various factors and the relationship diagram revealed that the influencing factors and test results could be described by a linear relationship [36]. It was assumed that the coal ash content was *x*_1_, the sodium silicate content was *x*_2_, the ettringite content was *x*_3_, the naphthalene sulfonate content was *x*_4_, the water–cement ratio was *x*_5_, the sand–cement ratio was *x*_6_, the compressive strength was *y*_1_, the bond stress was *y*_2_, and the adhesion stress was *y*_3_. The first 15 sets of orthogonal test data in Table 5 were substituted into Formulas (9)–(12), and regression analysis was performed to obtain the data shown in Table 12. Therefore, an empirical equation between *y* and *x* was obtained (Formula (13)).
(13)$$\left\{ \begin{array}{l} y_{1}=89.11-0.48x_{1}-2.47x_{2}+0.29x_{3}-0.89x_{4}-61.54x_{5}+0.8x_{6}\\ y_{2}=107.44+2.33x_{1}+2.84x_{2}+0.88x_{3}-2.87x_{4}-28.13x_{5}-7.65x_{6}\\ y_{3}=92.98+1.51x_{1}+5.11x_{2}+2.61x_{3}-1.49x_{4}-119.05x_{5}+6.14x_{6} \end{array}\right.$$

(2) Test verification of the model

The residual analysis on the model showed that all of the residual values were between the upper and lower limits of the confidence interval (Figure 13). This shows that the regression model was normal [37]. The latter three sets of data in Table 4 and the data presented in Table 10 were used as the test data for application with the obtained regression model to verify the rationality of the predictive results [38,39]. The specific values are shown in Table 13.

As shown in Table 13, the maximum relative error of the test data of the compressive strength was 6.81%. The maximum relative error of the bond stress test data was 7.42%. The maximum relative error of the adhesion stress test data was 2.44%. According to the “Standard for test method of basic properties of construction mortar” [40], it is reasonable to deviate the regression value from the experimental value by no more than 20%, indicating that the empirical equation could be used in application.

(3) Actual engineering forecast

In actual engineering applications, the parameters such as the length of the anchoring section, the diameter of the anchor, and the diameter of the borehole have a great influence on the safety and stability of the anchor. The bond stress *τ_b_* and the adhesion stress *τ_g_* could be obtained by Formulas (14) and (15) follows [25]:(14)$$\tau_{b}=\frac{1}{K}\times n\times \pi \times d_{s}\times \xi_{b}\times f_{b}\times L$$
(15)$$\tau_{g}=\frac{1}{K}\times \pi \times D\times f_{rb}\times L$$
where *n* and *d_s_* refer to the number of bars and diameter of the bolt (m), respectively, *K* is the operating condition coefficient, for which the permanent anchor was taken as 0.60 and the temporary anchor was taken as 0.72, *ξ_b_* is the bond strength reduction factor, taken as 0.85 when two steel bars are spot welded into a bundle, and as 0.85 and 0.70 when three steel bars are spot welded into a bundle, *L* is the anchor length (m), *f_b_* is the characteristic value of the bond strength between the bolt and anchorage body, as determined by the test, *D* is the anchor diameter (m), and *fr_b_* is the bond strength characteristic value between the anchorage body and rock layer and is determined by the test.

The bond stress in this study was a single bolt with the following characteristics: anchor length 0.6 m, bolt diameter 22 mm, *K* = 0.6, and *ξ_b_* = 1. Therefore, the bond stress *y*_2_ in the empirical equation could be expressed as:(16)$$y_{2}\approx \frac{1}{0.6}\times 1\times \pi \times 0.022\times 1\times f_{b}\times 0.6=0.022\pi f_{b}$$

After derivation, this could be written as:(17)$$\frac{\tau_{b}}{y_{2}}\approx \frac{1}{K}\frac{n\times \pi \times d_{s}\times \xi_{b}\times f_{b}\times L}{0.022\pi \times f_{b}}=\frac{1}{K}\frac{n\times d_{s}\times \xi_{b}\times L}{0.022}$$
(18)$$\tau_{b}\approx \frac{1}{K}\times \xi_{b}\times \frac{n\times d_{s}\times L}{0.022}\times y_{2}$$

The adhesion stress in this study was a test with an anchor length of 0.4 m and a bore diameter of 72 mm and *K* = 0.6. Therefore, the adhesion stress *y*_3_ in the empirical equation could be expressed as:(19)$$y_{3}\approx \frac{1}{0.6}\times \pi \times 0.072\times f_{rb}\times 0.4=0.048\pi f_{rb}$$

After derivation, this could be written as:(20)$$\frac{\tau_{g}}{y_{3}}\approx \frac{1}{K}\times \frac{\pi \times D\times f_{rb}\times L}{0.048\pi f_{rb}}=\frac{1}{K}\times \frac{D\times L}{0.048}$$
(21)$$\tau_{g}\approx \frac{1}{K}\frac{D\times L}{0.048}\times y_{3}$$

The above derivation analysis reveals that the regression model could effectively predict the bond stress and the adhesion stress in actual engineering, when the parameters such as the number of anchors, the length of the anchoring section, the diameter of the bolt, and the diameter of the borehole are determined. Only when the values of both stresses are greater than the design values of axial tension of bolt can the safety and stability of the anchoring system be ensured [25]. 

## 6. Application

Based on the deformation characteristics of the surrounding rock of the V# ore body return windway, the southward extension of the main road was selected for the support test. The support method shown in Figure 2 was chosen, the diameter of the bolt was 25 mm, and the diameter of the drilling hole was 72 mm. The cement anchoring agent material ratio was 12% for coal ash, 5% for sodium silicate, 6% for ettringite, 1.5% for naphthalene sulfonate, 0.4 for water–cement ratio, and 1.5 for sand–cement ratio. According to the stress of the monitoring site anchor, as shown in Figure 14. The design value of axial tension of bolt should be *Nt =* 325 kN. To ensure the stability of the anchoring system, *Nt < τ_b_* and *Nt < τ_g_* should be satisfied. Substituting these values into Equations (18) and (21) revealed that the anchoring section had a value of 1.5 m, which was 1/6 less than the length before the anchoring agent was optimized.

To verify the optimized cement anchoring agent material effect, mine pressure monitoring was carried out on the southward extension section of the V# ore body return windway, and two monitoring points were arranged. According to the field monitoring data, the roadway maximum deformations curve within the monitoring period could be obtained, as shown in Figure 15. After 90 days of monitoring, the maximum deformations of the two sides of the first measuring point was 51 mm, and the maximum deformations of the roof floor was 67 mm; the maximum deformations of the two sides at the second point were 59 mm; and the maximum deformations of the roof floor were 78 mm. The deformation values were all within the controllable range, the deformations tended to be stable at the later stages of monitoring, and the convergence rate was less than 1 mm/day. Therefore, the use of optimized anchorage material for surrounding rock support could effectively control large deformation of the roadway.

## 7. Conclusions

The following conclusions were obtained in this research:(1)The orthogonal design method was used to design the parameter proportioning scheme of different cement anchoring materials and the uniaxial compression test and pull-out force test were carried out. The uniaxial compressive strength of different test piece was 43–55 MPa, and the bond stress was 103–136 kN. The adhesion stress was 76–112 kN.(2)The sensitivity of each factor of anchor material to the physical and mechanical indexes of the specimen was analyzed by the range analysis method. The influence rule of various factors of the mechanical indexes of the specimen was expounded. Then, the analysis of variance method was used to analyze the significance of each factor to the physical and mechanical indexes of the test piece, and the correctness of the test results was verified. Finally, the optimal ratio of cement anchoring agent material was determined to be 12% coal ash, 5% sodium silicate, 6% ettringite, and 1.5% naphthalene sulfonate. The water–cement ratio was 0.4 and the sand–cement ratio was 1.5.(3)Through multivariate linear regression analysis, the empirical equations of the parameters of the anchoring agent material and the mechanical indexes of the test piece were obtained, and the regression model was tested and analyzed to verify the rationality of the prediction model. The results would be of great value in the future optimization of the anchoring agent materials and the application of engineering support design.(4)Field tests have shown that the use of optimized cement anchor material for surrounding rock support could effectively control the large deformation caused by high stress extrusion on the roof floor and two sides.

## Figures and Tables

**Figure 1 materials-13-00137-f001:**
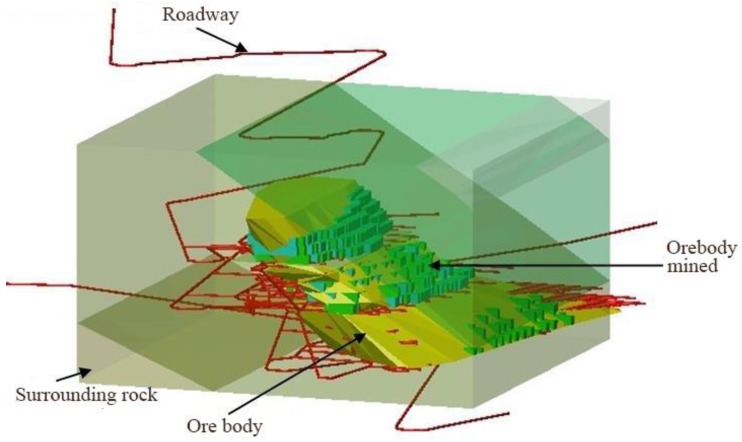
3D view of the V# ore body.

**Figure 2 materials-13-00137-f002:**
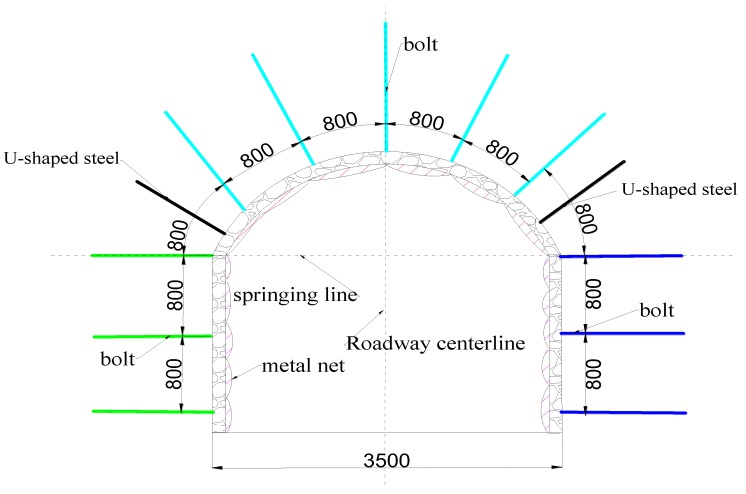
Roadway support.

**Figure 3 materials-13-00137-f003:**
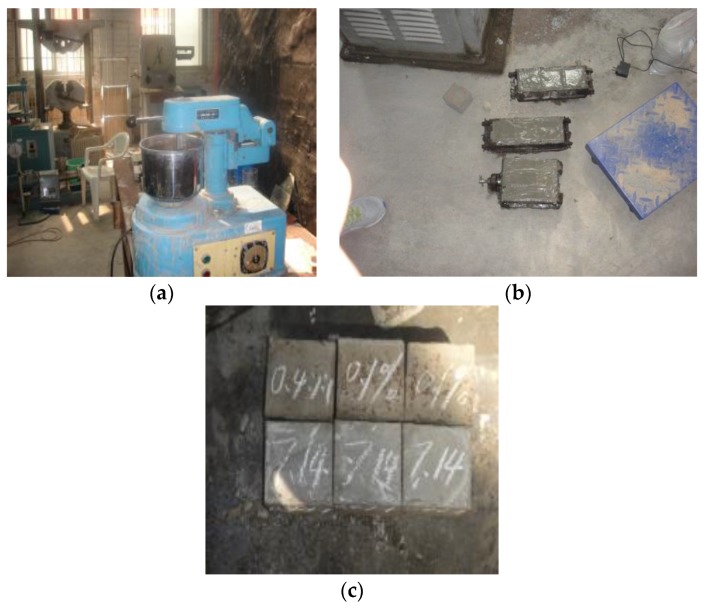
Preparation process of compressive strength test piece: (**a**) mixmaster; (**b**) test material preparation; (**c**) finished product.

**Figure 4 materials-13-00137-f004:**
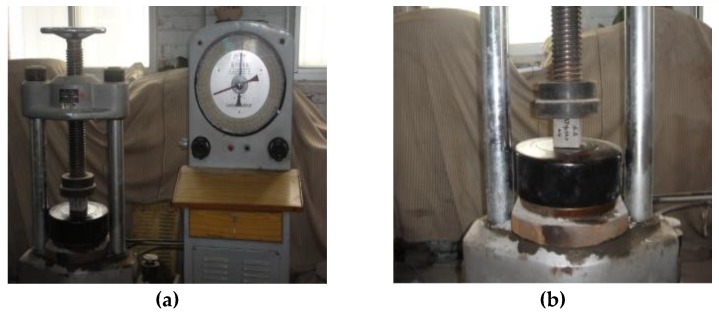
Compressive strength test: (**a**) compression testing machine; (**b**) test pieces of loading.

**Figure 5 materials-13-00137-f005:**
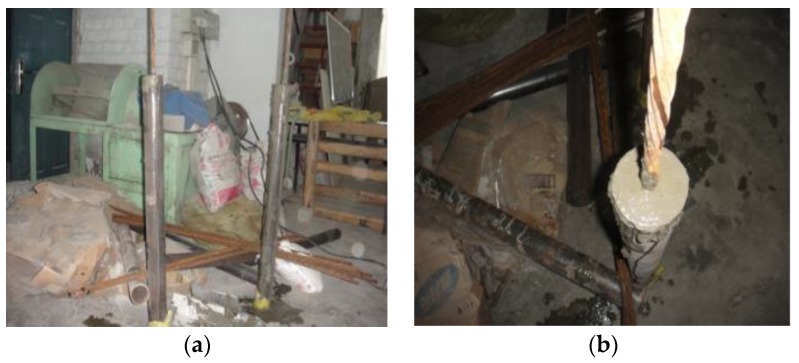
Bond stress test piece: (**a**) exterior; (**b**) interior.

**Figure 6 materials-13-00137-f006:**
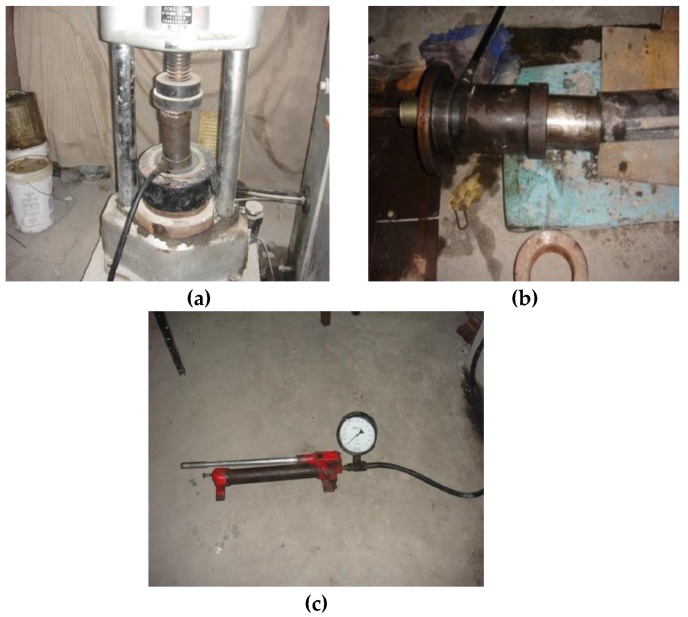
Anchor pull force test: (**a**) puller proofreading; (**b**) puller and test piece interface; (**c**) manual oil pump.

**Figure 7 materials-13-00137-f007:**
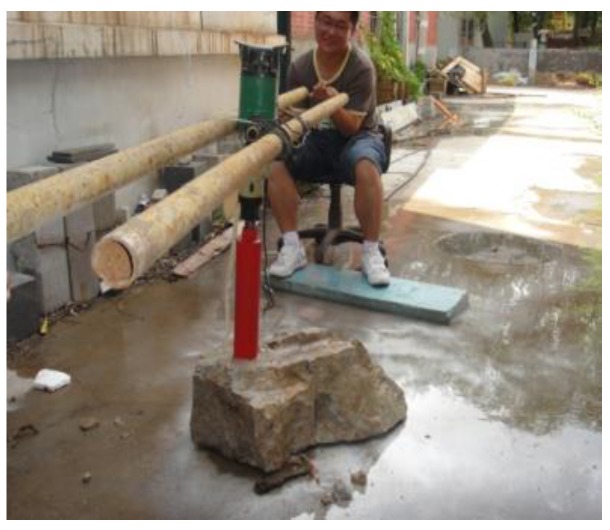
Drilling anchor hole.

**Figure 8 materials-13-00137-f008:**
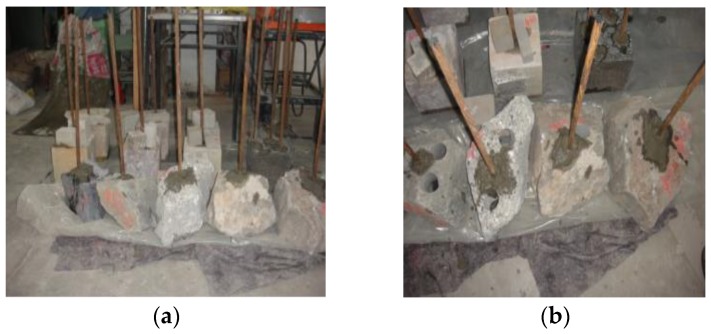
Adhesion stress test piece: (**a**) exterior; (**b**) interior.

**Figure 9 materials-13-00137-f009:**
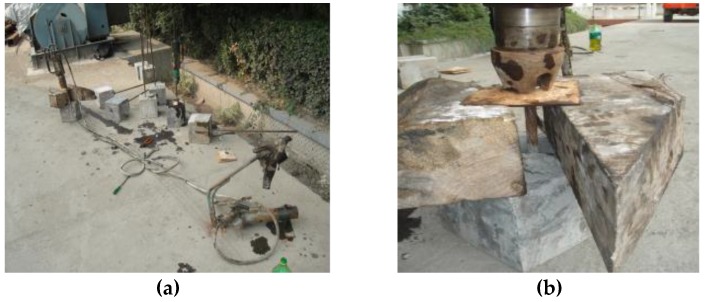
Adhesion stress test: (**a**) equipment installation; (**b**) test pieces of loading.

**Figure 10 materials-13-00137-f010:**
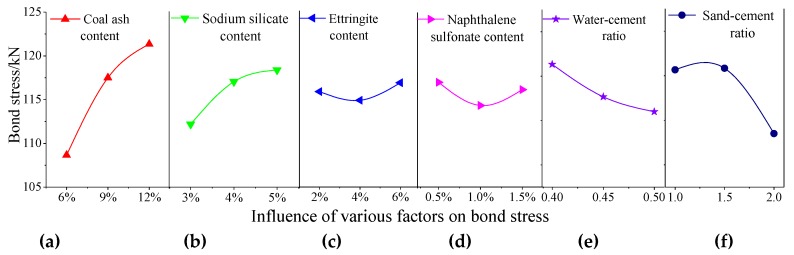
Trend diagram of the effect of various factors on the bond stress. (**a**) coal ash content; (**b**) sodium silicate content; (**c**) ettringite content; (**d**) naphthalene sulfonate content; (**e**) water–cement ratio; and (**f**) sand–cement ratio.

**Figure 11 materials-13-00137-f011:**
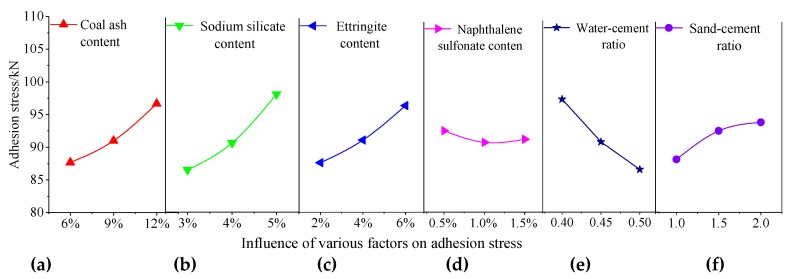
Trend diagram of the effect of various factors on the adhesion stress. (**a**) coal ash content; (**b**) sodium silicate content; (**c**) ettringite content; (**d**) naphthalene sulfonate content; (**e**) water–cement ratio; and (**f**) sand–cement ratio.

**Figure 12 materials-13-00137-f012:**
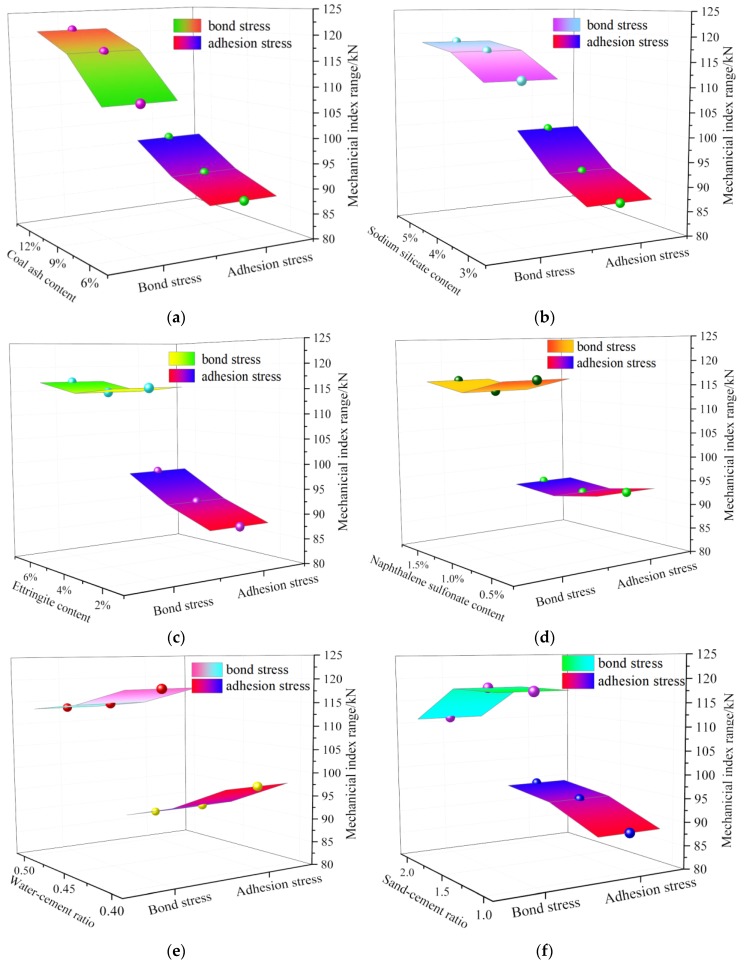
Comprehensive analysis of bond stress and adhesion stress. (**a**) coal ash content; (**b**) sodium silicate content; (**c**) ettringite content; (**d**) naphthalene sulfonate content; (**e**) water–cement ratio; and (**f**) sand–cement ratio.

**Figure 13 materials-13-00137-f013:**
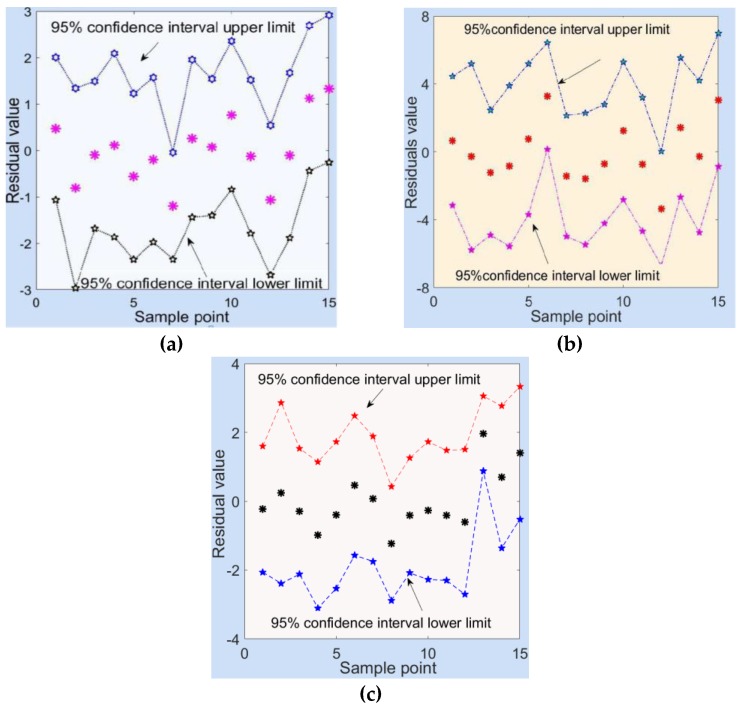
Residual value analysis. (**a**) Residual value of compressive strength; (**b**) residual value of the bond stress; and (**c**) residual value of the adhesion stress.

**Figure 14 materials-13-00137-f014:**
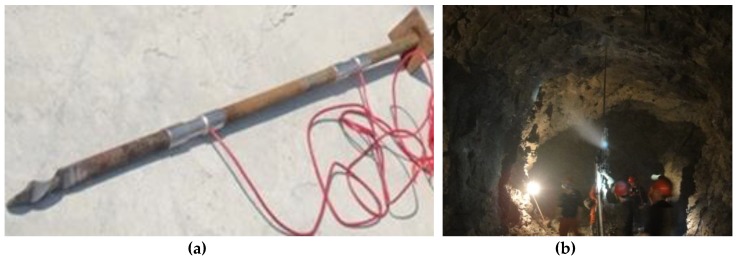
Monitoring of bolt stress. (**a**) Rock-bolt dynamometer and (**b**) arrangement of monitoring points.

**Figure 15 materials-13-00137-f015:**
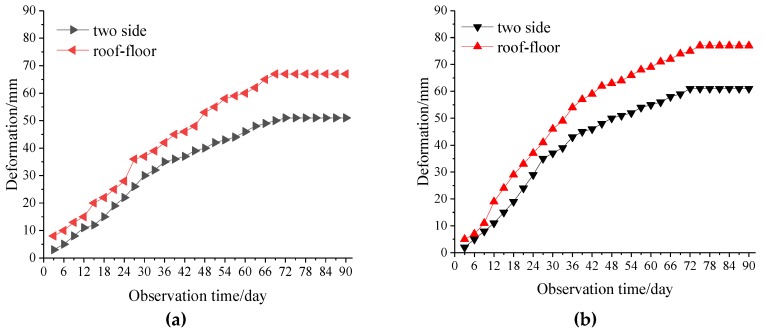
Roadway surrounding rock monitoring curve. Monitoring points (**a**) 1 and (**b**) 2.

**Table 1 materials-13-00137-t001:** Chemical composition of ordinary Portland cement.

Al_2_O_3_	CaO	SO_3_	SiO_2_	Fe_2_O_3_	MgO
6.65%	58.93%	2.54%	24.12%	3.78%	3.98%

**Table 2 materials-13-00137-t002:** Admixture performance.

Name	Ettringite (UEA)	Sodium silicate (Na_2_O·nSiO_2_)	Naphthalene sulfonate (NUF-5)	Coal Ash
Function	As the expansion agent, the volume of the anchored stone body of the grouting anchor can be increased to improve the ultimate pullout resistance of the anchor.	As a quick-setting agent, it reacts with the paste to form a gel with a certain strength, which improves the adhesion of the material.	As a water reducing agent, it increases the hydration efficiency and improves the durability of the cement.	As a blending material, the fluidity of the mixture can be improved, the density of the cement paste can be enhanced, and the bond force of the cement paste on the anchor rod can be improved.

**Table 3 materials-13-00137-t003:** Factors and levels of the orthogonal test.

Level	Coal Ash Content	Sodium Silicate Content	Ettringite Content	Naphthalene Sulfonate Content	Water– Cement Ratio	Sand– Cement Ratio	Blank Column
1	6%	3%	2%	0.5%	0.4	1	-
2	9%	4%	4%	1%	0.45	1.5	-
3	12%	5%	6%	1.5%	0.5	2	-

**Table 4 materials-13-00137-t004:** Anchoring agent material ratio scheme.

Test Number	Factor						
	Coal Ash Content	Sodium SilicateContent	Ettringite Content	Naphthalene SulfonateContent	Water– Cement Ratio	Sand– Cement Ratio	Blank Column
1	6%	3%	2%	0.5%	0.4	1	-
2	6%	4%	4%	1%	0.45	1.5	-
3	6%	5%	6%	1.5%	0.5	2	-
4	9%	3%	2%	1%	0.45	2	-
5	9%	4%	4%	1.5%	0.5	1	-
6	9%	5%	6%	0.5%	0.4	1.5	-
7	12%	3%	4%	0.5%	0.5	1.5	-
8	12%	4%	6%	1%	0.4	2	-
9	12%	5%	2%	1.5%	0.45	1	-
10	6%	3%	6%	1.5%	0.45	1.5	-
11	6%	4%	2%	0.5%	0.5	2	-
12	6%	5%	4%	1%	0.4	1	-
13	9%	3%	4%	1.5%	0.4	2	-
14	9%	4%	6%	0.5%	0.45	1	-
15	9%	5%	2%	1%	0.5	1.5	-
16	12%	3%	6%	1%	0.5	1	-
17	12%	4%	2%	1.5%	0.4	1.5	-
18	12%	5%	4%	0.5%	0.45	2	-

**Table 5 materials-13-00137-t005:** Orthogonal test results of anchoring agent material ratio.

Test Number	Mechanical Index		
	Compressive Strength/MPa	Bond Stress/kN	Adhesion Stress/kN
1	55	112	78
2	50	109	82
3	45	106	93
4	51	107	79
5	45	118	76
6	48	127	106
7	46	119	89
8	53	121	108
9	43	126	95
10	52	108	90
11	48	103	84
12	51	114	99
13	54	111	97
14	51	123	96
15	47	119	92
16	46	116	86
17	47	128	98
18	49	118	104

**Table 6 materials-13-00137-t006:** Range analysis of bond stress.

Number of Horizontal Groups	Coal Ash Content	Sodium Silicate Content	Ettringite Content	Naphthalene Sulfonate Content	Water–Cement Ratio	Sand–Cement Ratio	Blank Column
Average 1	108.667	112.167	115.833	117.000	118.833	118.167	116.000
Average 2	117.500	117.000	114.833	114.333	115.167	118.333	115.333
Average 3	121.333	118.333	116.833	116.167	113.500	111.000	116.167
Range	12.666	6.166	2.000	2.667	5.333	7.333	0.834

**Table 7 materials-13-00137-t007:** Analysis of variance of bond stress in the orthogonal test.

Factor	Deviation Sum of Squares	Degree of Freedom	F	F Critical-Value			Significance		
				a = 0.1	a = 0.05	a = 0.01	a = 0.1	a = 0.05	a = 0.01
Coal ash content	506.333	2	217.031	9	19	99	**	**	**
Sodium silicate content	126.333	2	54.150	9	19	99	**	**	*
Ettringite content	12.000	2	5.144	9	19	99	*	*	*
Naphthalene sulfonate content	22.333	2	9.573	9	19	99	**	*	*
Water–cement ratio	89.333	2	38.291	9	19	99	**	**	*
Sand–cement ratio	210.333	2	90.156	9	19	99	**	**	*
Error	2.33	2	-	-	-	-	-	-	-

“**” means that when the factor level changes, it has a significant effect on the test results;"*" means that when the factor level changes, it has a slight effect on the test results.

**Table 8 materials-13-00137-t008:** Range analysis of adhesion stress.

Number of Horizontal Groups	Coal Ash Content	Sodium Silicate Content	Ettringite Content	Naphthalene Sulfonate Content	Water–Cement Ratio	Sand– Cement Ratio	Blank Column
Average 1	87.667	86.500	87.667	92.833	97.667	88.333	91.333
Average 2	91.000	90.667	91.167	91.000	91.000	92.833	91.667
Average 3	96.667	98.167	96.500	91.500	86.667	94.167	92.333
Range	9.000	11.667	8.833	1.833	11.000	5.834	1.000

**Table 9 materials-13-00137-t009:** Analysis of variance of adhesion stress in the orthogonal test.

Factor	Deviation Sum of Squares	Degree of Freedom	F Ratio	F Critical-Value			Significance		
				a = 0.1	a = 0.05	a = 0.01	a = 0.1	a = 0.05	a = 0.01
Coal ash content	248.444	2	79.860	9	19	99	**	**	*
Sodium silicate content	419.444	2	134.826	9	19	99	**	**	**
Ettringite content	237.444	2	76.324	9	19	99	**	**	*
Naphthalene sulfonate content	10.778	2	3.464	9	19	99	*	*	*
Water–cement ratio	368.444	2	118.433	9	19	99	**	**	**
Sand–cement ratio	112.111	2	36.037	9	19	99	**	**	*
Error	3.111	2	-	-	-	-	-	-	-

“**” means that when the factor level changes, it has a significant effect on the test results;"*" means that when the factor level changes, it has a slight effect on the test results.

**Table 10 materials-13-00137-t010:** Optimal combination of orthogonal test factors and levels.

Group	Coal Ash Content	Sodium Silicate Content	Ettringite Content	Naphthalene Sulfonate Content	Water–Cement Ratio	Sand–Cement Ratio
Optimal parameter	12%	5%	6%	1.5%	0.4	1.5

**Table 11 materials-13-00137-t011:** Test results of mechanical indexes under optimal level combination.

Test Index	Compressive Strength/MPa	Bond Stress/kN	Adhesion Stress/kN
Test result	46	136	112

**Table 12 materials-13-00137-t012:** Test data fitting.

Index			Coefficient					*R* ^2^
	*b* _0_	*b* _1_	*b* _2_	*b* _3_	*b* _4_	*b* _5_	*b* _6_	
*y* _1_	89.11	−0.48	−2.47	0.29	−0.89	−61.54	0.8	92.1
*y* _2_	107.44	2.33	2.84	0.88	−2.87	−28.13	−7.65	94.5
*y* _3_	92.98	1.51	5.11	2.61	−1.49	−119.05	6.14	86.9

**Table 13 materials-13-00137-t013:** Error analysis of the fitting results.

Group Number	Compressive Strength/MPa			Bond Stress/kN			Adhesion Stress/kN		
	Trial Value	Regression Value	Relative Error/%	Trial Value	Regression Value	Relative Error/%	Trial Value	Regression Value	Relative Error/%
16	46.00	48.820	6.13	116	124.615	7.42	86	87.215	1.41
17	47.00	49.299	4.89	128	121.488	−5.09	98	96.115	−1.92
18	49.00	45.662	−6.81	118	123.726	4.85	104	105.05	1.44
Optimized group	46	48.389	6.15	136	127.848	−7.02	112	114.735	2.44

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
