# Peer review of "Experimental Study on Mix Proportion Parameter Optimization of Cement Anchoring Material"

_materials, 2019, doi:10.3390/ma13010137_

Round 1

Reviewer 1 Report

The manuscript needs some editorial and content corrections.

Abstract is not informative. That part of a paper should be a brief summary of the content. The obtained values for compressive strength, pull–out force the compressive strength of the test piece, the bond stress between the bolt and the anchoring agent, and the adhesion stress between the anchoring agent and the rock masses are strongly recommended to be added.

Experiments have been well designed and methods have been clearly presented. However, in discussion no references are given. In the present form it is a self-discussion. Refrences must be added, so that Authors discussed their own results against the literature. For example a statement in line 264: “Adding proper amount of ettringite to expand the volume of the slurry can increase the mechanical interaction force between the contact surfaces”. How do you know that? Did you examine expansion of sluury due to ettringite addition? I don’t think so. So add a reference.

References in the main text are in upper case. It is not correct.

DETAIL REMARKS:

Line 68 “above scholars” seems odd.

Line 74: “one the two bonding effects” is not correct. Suggest use “one of the two interfaces”

Line 172: “the size of” is unnecessary.

Line 184: lacking speed of compression (mm/min). Please add.

Line 203: lacking speed of pulling (mm/min). Please add.

Line 201: Was it done by pulling out the rod? Please explain and add speed of pulling.

Line 252-255: “The packing action of the reaction products of coal ash and cement slurry decreased the  macroporosity and capillary porosity in the cement stone while increasing the gel pores. The large change in the pore size distribution (large holes reduced, small holes increased) made the structure more compact and uniform”. How was it observed? In my opinion pictures would be helpful.

Table 6 & 8: please expalain what asterix stands for.

Regarding all above I recommend major revision.

Reviewer 2 Report

The paper concerns the investigations about the optimization of the anchoring mix technical parameters in terms of compressive strength, bond stress, and adhesion stress. The paper has the correct structure. The methodology is well described, results are presented clearly and were analyzed properly. However, in the paper there are some shortcomings, which should be corrected before acceptance the paper for publication. Detailed comments are listed below:

The style and the language should be improved. There are syntax and grammar errors. I suggest to check the paper by the English native speaker. In addition the paper should be checked by a specialist who is fluent in English, e.g., it is better to use the term "cement paste" than "cement slurry". Please make sure the paper's formatting complies with the journal's standard. If possible, please provide the basic characteristics of the cement used, i.e., chemical composition, specific surface area, etc. Line 178 - why the samples for determining the compressive strength have 70.7x70.7x70.7 mm dimensions? This is quite a rare sample size for this type of test. Table 4 - Please provide the number of samples in one test and the basic measure of the results dispersion, e.g., standard deviation or coefficient of variation. Without these data it is not possible to determine the repeatability or reliability of the results obtained. The authors should add that practically all the mechanical characteristics of the cement paste deteriorates with the increase of the water-cement ratio, which is crucial in the case of the anchorage systems. We are talking here about such parameters as: compressive strength, tensile strength, shrinkage deformations, crackability, thermal resistance, fatigue resistance, etc. Examples of references in the literature in this topic:

"Mechanical properties and microstructure of multi-walled carbon nanotube-reinforced cement paste." Construction and Building Materials 76 (2015): 16-23.

"Properties of cracking patterns of multi-walled carbon nanotube-reinforced cement matrix." Materials 12.18 (2019): 2942.

Reviewer 3 Report

It is an interesting article presenting the ratio optimization of cement anchoring material based on the orthogonal design method and rock mechanics tests. The authors should address the following contents to improve the quality of the paper:

Line 15: The words “and” is missing from the following sentence in the abstract:

Compressive strength and pull–out force tests were carried out….etc.

Line 19: You cannot use the same word “analyze/ analysis” three times in one sentence.

Please try to improve the writing quality throughout the whole manuscript.

Round 2

Reviewer 1 Report

The clarity of manuscript after revision is improved. All the lacks indicated in a revision form have been addressed. The paper can be published in the present form.

Reviewer 2 Report

The suggestions have been included, the paper can be published.